# Genetic and pharmacological interrogation of cancer vulnerability using a multiplexed cell line screening platform

Yifeng Xia [1✉], Xiaodong Ji [1✉], In Sock Jang[1], Christine Surka[1], Christy Hsu[1], Kai Wang [1], Mark Rolfe[1], Neil Bence[1] & Gang Lu[1]

The multiplexed cancer cell line screening platform PRISM demonstrated its utility in testing hundreds of cell lines in a single run, possessing the potential to speed up anti-cancer drug discovery, validation and optimization. Here we described the development and implementation of a next-generation PRISM platform combining Clustered Regularly Interspaced Short Palindromic Repeats (CRISPR)/Cas9-mediated gene editing, cell line DNA barcoding and next-generation sequencing to enable genetic and/or pharmacological assessment of target addiction in hundreds of cell lines simultaneously. Both compound and CRISPR-knockout PRISM screens well recapitulated the results from individual assays and showed high consistency with a public database.

[1] Bristol Myers Squibb, San Diego, CA, USA. ✉email: yifengxia@yahoo.com; xiaodong.ji@bms.com

Human cancer cell lines serve as a powerful tool for defining the mechanism of cancer growth and metastasis, as well as for developing therapeutic interventions to attack cancer vulnerabilities. More than 1000 cancer cell lines from various tumor types have been established in the past decades and are widely used by the cancer research community[1]. However, these cancer cell lines, which are passaged for many years in culture dishes, may no longer fully recapitulate their original genetic and/or epigenetic characteristics owing to clonal selection[2,3]. It is not rare that scientific findings from one laboratory cannot be reproduced by other researchers using different strains of the same cell line that underwent genomic evolution. As such, a better approach to faithfully capture cancer vulnerability specific to a tissue lineage or genetic background is to assess the response to inactivation of a candidate disease-driving protein across a large number of cell lines covering common and distinctive genetic characteristics. This practice is usually impractical in small laboratories with limited resources, thus calling for the need for a reliable platform that allows the screening of hundreds of cell lines in a high-throughput manner.

Broad Institute's Project Achilles through the DepMap portal offers the assessment of gene essentiality for the majority of protein-coding genes in >700 human cancer cell lines based on Clustered Regularly Interspaced Short Palindromic Repeats (CRISPR)/Cas9- as well as RNA interference (RNAi)-mediated gene perturbation[4,5]. The dependence on individual genes was determined using a predefined algorithm in cancer cell lines under standard nutrient-rich culture conditions for 10–12 cell doublings, resulting in the identification of potential therapeutic targets for various cancer indications. Although CRISPR/Cas9-based gene knockout has proven to be a powerful tool for inactivating gene function with greatly improved precision over traditional RNAi-mediated knockdown[6,7], follow-up validation studies are needed to confirm whether knockout of an indicated target gene leads to loss of cell fitness via an on-target mechanism in a panel of cell lines. This effort would still take a large amount of time and resources if performed in dozens of cell lines individually.

In 2016, researchers from the Broad Institute reported the development of PRISM—a multiplexed cell line screening platform combining a DNA-barcoding technique and a Luminex microsphere detection system[8]. The PRISM technology enabled the profiling of compound activity on cell viability in dozens to hundreds of cell lines in a single run to define genotype-specific cancer vulnerabilities. Recently, PRISM was successfully applied to evaluate the growth-inhibitory activity of 4518 drugs across 578 human cancer cell lines, revealing the potential application of non-oncology drugs for the treatment of cancer[9]. However, the PRISM platform is not readily amenable for high-throughput interrogation of cancer vulnerability using genetic tools, such as CRISPR/Cas9-mediated gene editing.

In order to enable the rapid validation of the therapeutic value of a given target of interest across a large panel of human cancer cell lines using both CRISPR and pharmacological perturbations, we developed BMS (Bristol Myers Squibb)-PRISM platform combining CRISPR/Cas9-mediated gene editing capability and DNA-barcoding multiplexing technique. We carried out a focused run with epidermal growth factor receptor (EGFR) and a full library run with KRAS as proof-of-principle studies. Both compound and CRISPR-knockout PRISM screens well recapitulated the results from individual assays and showed high consistency with public database.

## Results

### Development of BMS-PRISM platform

We designed a next-generation PRISM screen platform that incorporates DNA-barcoding technology and CRISPR/Cas9-mediated gene editing (Fig. 1). We first generated a BMS-PRISM plasmid library of lentiviral constructs that each expresses *Streptococcus pyogenes*-derived Cas9 (*spCas9*) and a blasticidin-resistant gene (*blast*), with a unique 26-base pair (bp) DNA barcode between them. We integrated these constructs into cell lines of interest via lentiviral transduction and then selected with blasticidin to establish pooled stable lines. The Cas9 editing efficiency was then validated using a lentiviral Cas9 activity reporter (Supplementary Fig. 1). Over the past 2 years, we have individually engineered >400 solid tumor cell lines obtained from ATCC, JCRB, DSMZ, and ECACC, covering most major cancer types. Only cell lines with Cas9 editing efficiency >85% were archived into the BMS-PRISM collection (current version = 368 lines).

We then developed a next-generation sequencing (NGS) protocol for quantification of the percentage of each unique barcode in a mixture, which should faithfully represent the relative cell number of the corresponding cell line in a mixed pool with or without genetic and/or pharmacological manipulation. In brief, the barcode regions integrated into the genomic DNA were PCR amplified with a pair of universal primers. A second PCR reaction was performed on the first PCR product to incorporate dual-indexed Illumina primers into the final barcode library. The abundance of unique barcodes in each library was quantified via NGS (Supplementary Fig. 2a).

To determine the linear range of the NGS method for measuring the abundance of unique 26-bp DNA barcodes with potentially variable representations in a NGS library, 20 lentiviral constructs with Cas9/barcodes were randomly selected and diluted into five concentration groups (100, 10, 1, 0.1, and 0.01 ng/μl), with each group containing 4 barcoded plasmids. Next, equal volumes of these diluents were mixed to generate the test plasmid pool, in which the relative representation of each of the 20 unique barcodes was quantified using the NGS protocol. As shown in Supplementary Fig. 2b, all 20 barcodes were successfully detected at the expected relative molarity, supporting that this NGS method can detect barcodes quantitatively with a 10,000-fold difference in their abundance in the library.

### Focused screen with EGFR perturbation

To optimize the screening protocol for assessing cell lines' sensitivity to pharmacological inhibition or genetic inactivation via CRISPR/Ca9-mediated gene editing, we selected 20 non-small cell lung carcinoma (NSCLC) cell lines with wild-type or mutant EGFR in the BMS-PRISM collection (Supplementary Table 1). Oncogenic addiction to EGFR mutations is commonly found in NSCLC patients and EGFR-mutated cell lines are vulnerable to genetic knockout or selective receptor tyrosine kinase inhibitors, such as Erlotinib (Tarceva)[10]. To establish the EGFR dependence of these 20 NSCLC cell lines, we first evaluated the sensitivity of each cell line to Erlotinib treatment (CellTiter-Glo (CTG) cell viability assay) or EGFR knockout (flow cytometry-based CRISPR competition assay). HCC827 (E746-A750del), HCC4006 (L747-E749del, A750P), and PC9 (E746-A750del) cells harboring typical EGFR exon 19del mutations were most sensitive to Erlotinib or EGFR knockout, while others showed little to no response to EGFR perturbation (Fig. 2a, b and Supplementary Fig. 3). In the PRISM screen (detailed protocol described in Supplementary Methods), we also spiked two barcoded (Cas9-dead) 293T control cell lines into the cell line mix and used them as internal references to normalize relative abundance. The cell fitness of 293T cells was not affected by most of the perturbations tested so far, although Erlotinib at 3 and 10 μM did elicit potent cytotoxicity (Fig. 2a). Therefore, we adjusted the barcode abundance of 293T in the PRISM cell line mix according to the cell fitness of 293T cells under the same treatment condition in

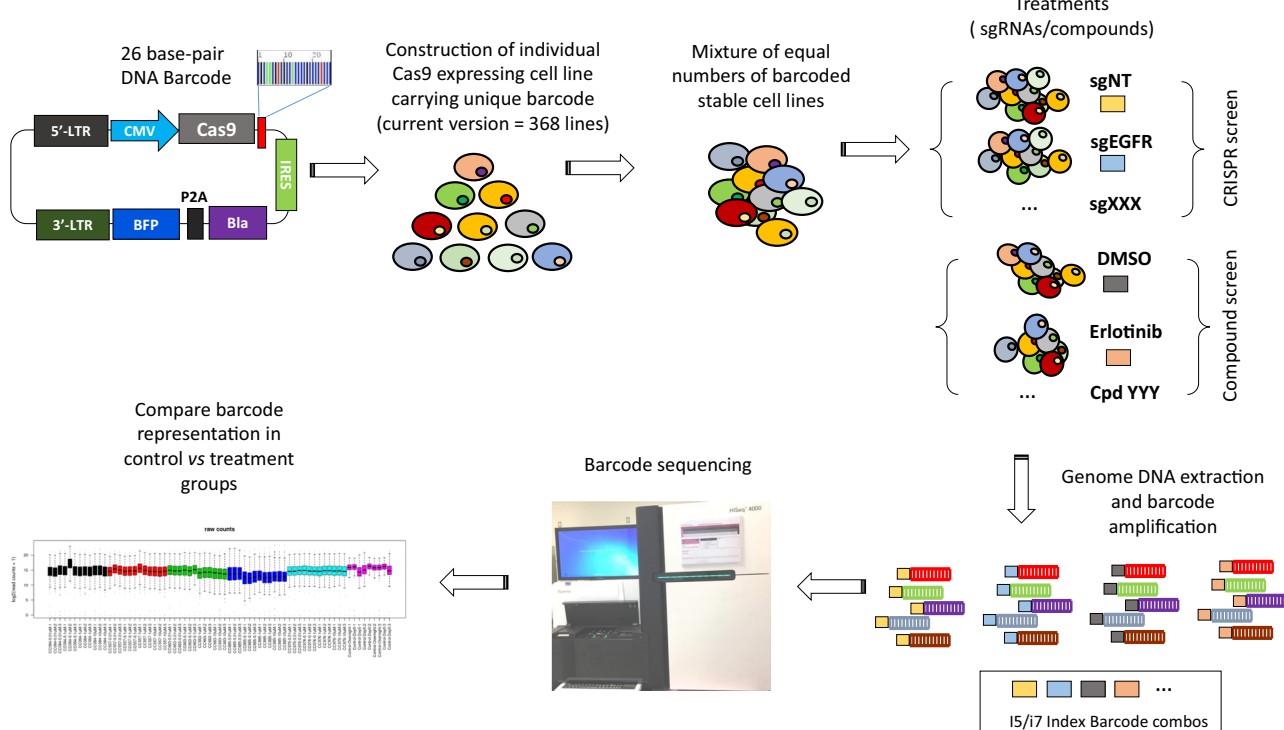

**Fig. 1 Overview of BMS-PRISM platform.** Solid tumor cell lines were individually infected with Cas9 and 26-bp barcode and tested for CRISPR editing efficiency before archived into BMS-PRISM library. These engineered cell lines were then subpooled and banked according to cancer indications for convenient handling. A full or focused collection of cell lines can be used for the compound as well as CRISPR/Cas9 knockout screening to evaluate drug response or gene essentiality in a single run. Relative cell line abundance after various treatments can be determined by PCR amplification of barcodes and next-generation sequencing.

the 5-day CTG assay, and the relative 293T barcode abundance was then used to calculate the barcode representation of each cell line in the same PRISM mix (Supplementary Fig. 4). After this adjustment, all cell lines demonstrated a good correlation between the AUC (area under the dose–response curve) of Erlotinib assessed in the PRISM screen and the AUC of the individual CTG assay carried out in-house or reported in the Cancer Therapeutic Response Portal (Fig. 2c, d). Consistently, the EGFR gene essentiality score in the majority of cell lines is correlated to that of the individual CRISPR competition assay aforementioned, as well as to the DepMap CRISPR essentiality score (CERES, Fig. 2e, f). Small differences in compound sensitivity or gene essentiality between individual tests and the pooled PRISM screen could be attributed to differences in cell culture conditions as discussed in detail below.

**Full-library screen with KRAS perturbation**. KRAS mutations are found in nearly one-third of all human malignancies worldwide, among which KRAS G12C is highly smoking related and can now be selectively targeted by covalent inhibitors, such as AMG-510, which is currently under clinical investigation[11,12]. Since both KRAS dependency and AMG-510 specificity are well defined, we picked KRAS as one of our targets to validate the BMS-PRISM platform. Out of the total 17 KRAS G12C mutant cell lines in the current BMS-PRISM collection, most showed decent AMG-510 sensitivity in the PRISM compound screen, while 61 cell lines with non-G12C KRAS mutations did not respond to AMG-510 treatment (Fig. 3a, b). In comparison, most cell lines with either G12C or non-G12C KRAS mutations showed great depletion in the KRAS CRISPR knockout screen. In order to further demonstrate the reproducibility of the PRISM

platform, we chose four KRAS-G12C, four KRAS non-G12C mutant, and three KRAS wild-type cell lines for individual validation of their response to AMG-510 treatment or KRAS CRISPR knockout. As shown in Fig. 3d, KRAS wild-type cell lines did not respond to AMG-510 or KRAS knockout, while non-G12C KRAS mutant cells only responded to KRAS knockout but not to AMG-510. Interestingly, among the group of G12C cell lines, SW1573 showed very little response to either AMG-510 or KRAS knockout, which was consistent with earlier reports that some of the KRAS mutant cells do not show KRAS dependency under conventional two-dimensional (2-D) cell culture conditions[13,14]. Notably, LU99 cells responded well to KRAS knockout in both the individual assay and the PRISM screen but only showed a mild response to AMG-510 treatment. It is possible that this cell line harbors a certain resistance mechanism to the compound but still maintains KRAS dependency. In contrast, SW837, another G12C cell line, showed KRAS dependency to both AMG-510 and KRAS knockout in the individual culture conditions but not in the mixed PRISM culture condition. This is likely owing to the aforementioned non-cell-autonomous paracrine effect in the mixed cell culture that may compensate for the KRAS dependency in supporting cell proliferation. This effect should be investigated further particularly when gene dependency relies heavily on culture conditions (e.g., 2-D vs three-dimensional (3-D) cell culture). Nevertheless, most of the 17 KRAS G12C mutant cell lines showed their consistent response to AMG-510 and KRAS CRISPR PRISM screen (Supplementary Fig. 5a). Finally, our KRAS CRISPR PRISM screen result correlated well with what has been reported at DepMap portal (Fig. 3e). KRAS mutant cancer cell lines were relatively more sensitive to KRAS genetic knockout. Overall, the KRAS screen further demonstrated the

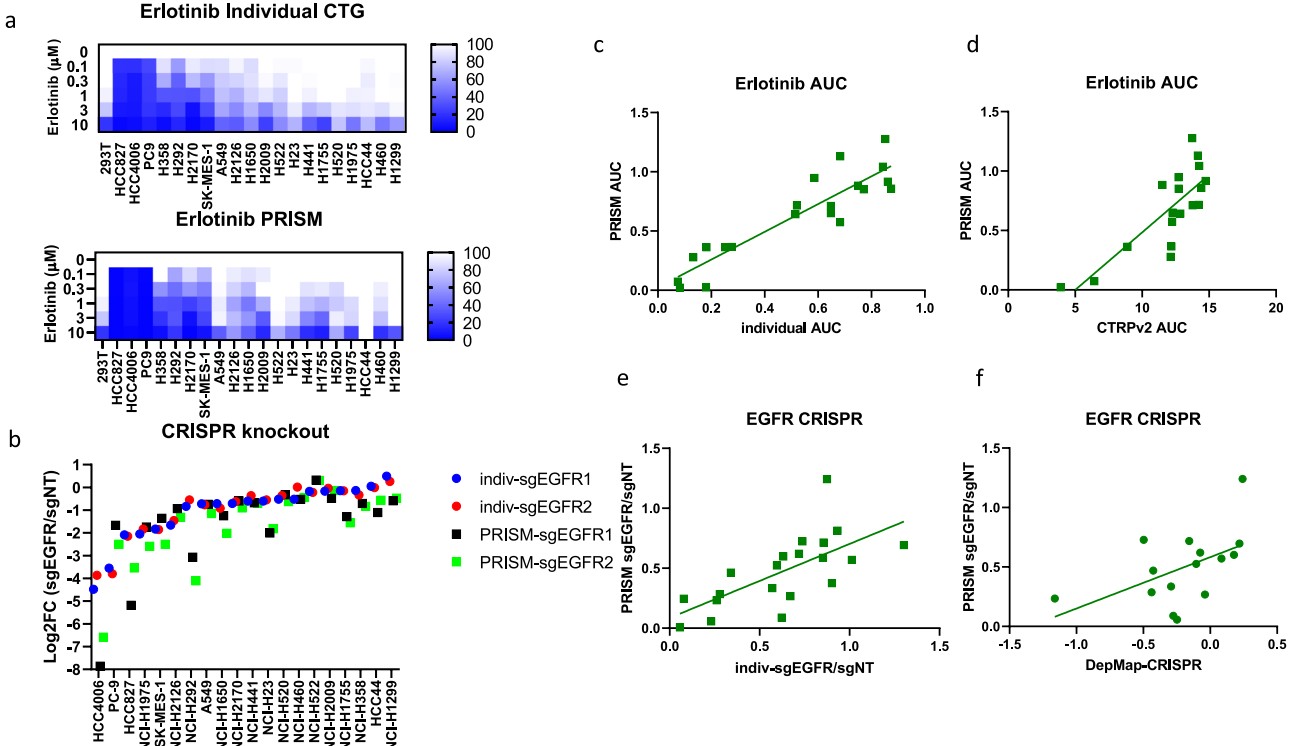

**Fig. 2 EGFR-dependency evaluation in a focused PRISM analysis of 20 NSCLC lines. a** Heatmap showed the response of 20 NSCLC lines to 5-day Erlotinib treatment (0, 0.1, 0.3, 1, 3, 10 μM) in individual CellTiter-Glo assay (top) or PRISM assay (bottom). **b** EGFR gene dependency of 20 NSCLC lines was evaluated by CRISPR competition assay (7 day, solid circles) and PRISM assay (5 day, solid squares). EGFR gene was knocked out by two single guide RNAs (sgEGFR1 and sgEGFR2), respectively. **c** Correlation analysis of Erlotinib AUC in individual CTG assay vs PRISM assay (20 cell lines, r = 0.9097). **d** Correlation analysis of Erlotinib AUC in PRISM assay vs AUC reported by Cancer Therapeutics Response Portal (CTRPv2, 18 cell lines, r = 0.7969). **e** Correlation analysis of EGFR CRISPR competition assay vs PRISM assay (plotted with average of two sgRNAs, 20 cell lines, r = 0.6844). **f** Correlation analysis of EGFR CRISPR PRISM assay vs DepMap gene effect score (CERES, 15 cell lines, r = 0.5098).

utility of the BMS-PRISM platform in evaluating the response of multiple cell lines in a single mixture to compound treatment or genetic manipulation, with the cell lines responding to AMG-510 or KRAS knockout as expected based on their KRAS mutation status.

**An example of cereblon (CRBN) modulator screen**. CRBN-mediated GSPT1 degradation by the next-generation CRBN modulator CC-885 has been shown to have a strong anti-proliferative effect in most cancer cell lines with adequate CRBN expression and activity[15]. We therefore included CC-885 in the validation of the BMS-PRISM compound screen. Unlike AMG-510, which only had an antiproliferative effect in a few cell lines harboring the KRAS G12C mutation, CC-885 showed strong activity in most of the cell lines, including the 293T cells which we usually spike into the PRISM mix as an internal control. To address this problem, we spiked two 293T CRBN knockout cell lines into the PRISM mix as an internal control for read normalization in this CC-885 screen. As expected, the compound AUC calculation correlated well to CRBN mRNA level for most of the cell lines in the PRISM mix; the cell lines with lower CRBN expression tended to be more resistant to CC-885 treatment (Supplementary Fig. 6a).

CRBN-mediated protein degradation is one of the most attractive strategies in the drug discovery field in recent years[16]. Interestingly, CRBN expression and activity differs widely among different tissues, which is one of the key factors to consider when exploring protein degradation targets. For example, colorectal and gastric cancers have relatively high CRBN expression and

activity (Supplementary Fig. 6b), signifying their potential as top cancer indications to consider when developing CRBN-mediated protein degradation strategies.

## Discussion

Cell line models have been actively serving cancer biology and drug discovery for many years despite their apparent limitations, such as their poor representation of heterogeneity, a key feature of human cancer. However, this concern can be largely addressed if one uses a collection of hundreds of cancer cell lines to draw a target prediction and test a therapeutic hypothesis. Although it was technically difficult to carry out such large-scale experiments individually, it now becomes possible when taking advantage of DNA-barcoding and next-generation sequencing technologies with our BMS-PRISM platform. Furthermore, BMS-PRISM has CRISPR/Cas9 knockout capability, a new feature of the cell line collection that was absent from the Broad-PRISM platform published in 2016. In this study, we used 20 NSCLC lines in a pilot screen to optimize the biology set-up and data-processing pipeline of the BMS-PRISM platform. Subsequently, we have successfully demonstrated the robustness and reproducibility of this platform in a compound screen as well as a CRISPR knockout screen with 368 cell lines covering all major solid tumor indications. Overall, the results derived from the PRISM screen were very consistent with those derived from assays run individually on each cell line.

One of the major differences between the PRISM screen and individual assays that may introduce inconsistency is the cell culture condition. First, nearly 400 cell lines were mixed together

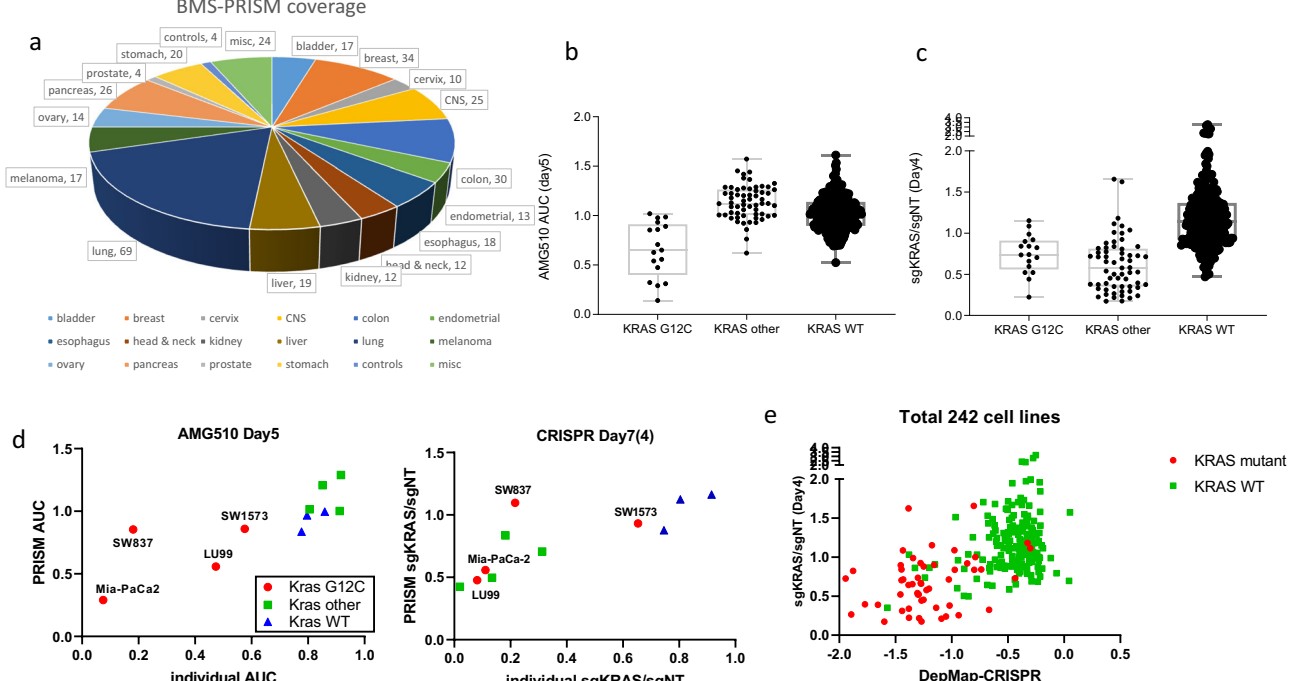

**Fig. 3 KRAS-dependency evaluation in full-library PRISM analysis. a** Pie chart showed the coverage of BMS-PRISM library composition. Total 368 cell lines were included in the current study. **b** Cell line response to Kras G12C inhibitor AMG-510 in PRISM screen. Cell lines were divided into three groups: KRAS G12C mutant (17 lines), KRAS other mutant (non-G12C, 61 lines), and KRAS wild-type (290 lines). **c** Cell line response to KRAS CRISPR knockout in PRISM screen (plotted with average of two sgRNAs). Cell lines were divided into three groups: KRAS G12C mutant (17 lines), KRAS other mutant (non-G12C, 61 lines), and KRAS wild-type (290 lines). Error bars represent standard error of the mean. **d** Validation of cell line response to KRAS perturbation in individual and PRISM assays (left, AMG-510 treatment; right, KRAS CRISPR knockout). KRAS G12C mutant (red circles); KRAS non-G12C mutant (green squares); KRAS wild-type (blue triangles). **e** Correlation analysis of KRAS CRISPR PRISM assay vs DepMap CRISPR gene effect score (CERES, 242 cell lines). KRAS mutant (red circles); KRAS wild-type (green squares).

and cultured in a single pool for 5–10 days in the PRISM screen settings, and therefore paracrine signal(s) may affect the dependence on a gene of interest in certain cell populations. Although it is conceivable that the paracrine effects of genes involved in regulation of cell proliferation signals, such as EGFR and KRAS, should become more dominant in the PRISM screen, we observed minimal discrepancy between the PRISM screens and their individual assays when EGFR and KRAS were tested in a focused screen format and full-library screen format, respectively. We reasoned that the percentage of cell lines that could potentially affect the cell fitness of others via a paracrine effect controlled by any gene of interest in the PRISM screen is probably <5%, thus this paracrine effect, if existed, should not affect the overall scoring of gene dependency in the PRISM screen. This hypothesis is strongly supported by our data shown in this study as well as in the PRISM screen result published by the Broad Institute[8,9]. On the other hand, given the limited cell number of each cell line (roughly 1000 cells/per line) in the PRISM screen pool, the autocrine effects observed in cell lines when cultured as a homogenous population may not be detected in the PRISM screen, which could lead to differential addiction to genes affecting cell growth via an autocrine signal. Lastly, cell seeding density is another plausible factor affecting the readout of gene dependency in the PRISM screen. Notably, cell density is relatively higher in the PRISM screen than in standard CTG or CRISPR competition assays, so cell–cell and cell–matrix interactions may vary slightly from traditional cell culture conditions. Maintaining the cells in a monolayer in the PRISM screen helps to overcome this limitation, since cells may acquire some features

of 3-D culture conditions once they outgrow the monolayer. For example, since we were able to maintain cells in a monolayer in the PRISM setting, we did not observe increased dependency for KRAS of any cell lines, which has previously been shown under 3-D culture conditions. Nevertheless, further investigation is needed to assess the difference in cell culture condition on gene dependency, especially in cell lines showing differential addiction to a given gene of interest. Moreover, since tumor cells in a human body typically grow in a nutrient-limited microenvironment where complex autocrine and paracrine cell–cell and/or cell–matrix interactions control the tumor growth and survival, the true dependence on any candidate gene needs to be validated in a panel of cell line or patient-derived xenograft models in vivo.

It is worth mentioning that normalization of raw reads is critical for the PRISM platform data interpretation, as hundreds of cell lines were cultured together as a mix and their proliferation rate varies. In BMS-PRISM, we did not quantify the barcode copy number of each cell line so the initial abundance of each barcode is not equal. For normalization purpose, we collect control samples (dimethyl sulfoxide (DMSO) treated or sgNT (non-targeting control single-guide RNA (sgRNA)) infected) at every time point as references. The abundance of each barcode in the treatment groups are normalized by that in the control group. For CRISPR screen, there is another critical factor that needs to be considered—different cell line may have different sensitivity to Cas9-induced DNA damage response (double-strand breaks). Therefore, we always include two control sgRNAs: sgNT and sgNC. sgNT does not match any genomic sequence so it does not introduce double-strand breaks, while sgNC matches the non-

coding sequence (intron) of HBG1 gene. HBG1 is only expressed in the fetal liver, spleen, and bone marrow, so targeting its intron sequence in cancer cell lines is expected to have minimal on-target effect. Therefore, sgNC serves as a control for cell line's sensitivity to DNA damage response. Interestingly, we did not observe apparent change of cell abundance upon the infection of sgNT and sgNC in most cell lines (Supplementary Fig. 5b). This result suggested that DNA damage response upon Cas9 editing does not impair cell fitness.

Altogether, the BMS-PRISM platform has demonstrated its high-throughput capacity, high flexibility, and high reproducibility in both compound screen and CRISPR knockout screen settings and therefore will greatly speed up oncology target validation and compound optimization activities, allowing for the screening of hundreds of cancer cell lines simultaneously.

## Methods

**Cell culture, lentiviral vectors, and chemicals**. Human embryonic kidney cell lines 293T and all solid tumor cell lines were purchased from ATCC, JCRB, DSMZ or ECACC and maintained in Dulbecco's Modified Eagle's medium (Invitrogen) or RPMI-1640 medium (Invitrogen) supplemented with 10% fetal bovine serum (Invitrogen), 1× sodium pyruvate (Invitrogen), 1× non-essential amino acids (Invitrogen), 100 U/ml penicillin (Invitrogen), and 100 μg/ml streptomycin (Invitrogen). PRISM cell mix was cultured in complete RPMI-1640 medium as described above. Lentiviral vectors expressing Cas9 and unique 26-bp DNA barcode or Cas9 activity reporter were custom-synthesized at Genscript. EGFR inhibitor Erlotinib and KRAS G12C inhibitor Sotorasib (AMG-510) were purchased from Selleckchem.

**Cell proliferation assay (CTG)**. Human cancer cell lines cultured in the complete growth medium were seeded into 96-well plates containing DMSO or test compounds. The seeding density for each cell line (typically 2000–3000 cells per well) was optimized to allow the cell growth in the linear range during culture period (5–7 days). After the culture period, cell proliferation was assessed using CTG Luminescent Cell Viability Assay (Promega) according to the manufacturer's instructions. Individual compound test AUC was calculated with 10-point CTG assay data (0–10 μM, half-logarithmic dilution).

**CRISPR-based competition assay**. Gene essentiality was evaluated with CRISPR-based competition assay. Briefly, human cancer cells were infected with red fluorescent protein (RFP)-sgNT or RFP-targeting sgRNA (two sgRNAs per gene) and then mixed 1:1 with cells infected with green fluorescent protein (GFP)-sgNT on Day3 (third day post lentiviral infection). The ratios of RFP/GFP cells were monitored on Day3, Day7, Day10, and Day14. RFP-sgTargeting/GFP-NT ratio normalized by RFP-sgNT/GFP-sgNT (typically on Day10) was used to evaluate the gene essentiality in this human cancer cell line.

The sgRNA sequences used in this study were: sgNT (non-targeting, GTAGCGAACGTGTCCGGCGT); sgNC (targeting HBG1 non-coding region, GGCCAGTGACTAGTGCTTGA); sgEGFR (1# CTGCGCTCTGCCCGGCGAGT; 2# TGCAAATAAAACCGGACTGA); and sgKRAS (1# AAGAGGAGTACAGTGCAATG; 2# AGATATTCACCATTATAGGT).

**BMS-PRISM platform**. Detailed screen protocol is described in Supplementary Methods and Supplementary Fig. 7.

**Genomic DNA isolation and sequencing library preparation**. To generate PRISM libraries for NGS, genomic DNA was isolated from up to $5 \times 10^6$ cells from each sample consisting of mixed PRISM cell lines using the QIAamp DNA Mini Kit (Qiagen) according to the manufacturer's instructions, with an RNase A treatment added upon cell lysis to eliminate cellular RNA. Total genomic DNA was visualized and quantified on the TapeStation 4200 using the Genomic DNA tape (Agilent). To amplify the 26 bp DNA barcode portion of the Cas9 construct transduced into each cell line, PCR was performed on 1 μg of genomic DNA per sample using custom primers that flanked the barcode region (Supplementary Fig. 2a) with sequences 5′-ACAACAAGCACCGGGATAAG-3′ (forward primer) and 5′-AGGAACTGCTTCCTTCACGA-3′ (reverse primer) and the Titanium Taq Polymerase and PCR Kit (Takara). Twenty-four cycles of PCR were performed with an annealing temperature of 65 °C and the 550 bp PCR products were visualized and confirmed on a 2% agarose gel. The 100 μl PCR reactions were then cleaned with 1× volumes of SPRIselect beads (Agencourt) to eliminate primers according to the manufacturer's instructions. The cleaned products were eluted in Elution Buffer (Qiagen) and quantified on the TapeStation 4200 with the High Sensitivity D1000 DNA tape (Agilent). To generate the final sequencing libraries, a second PCR reaction was performed on 1 ng of each first PCR product to incorporate dual-indexed Illumina primers each containing unique 8-nucleotide indexes. Six cycles of PCR were run with an annealing temperature of 65 °C and six

more cycles were run with an annealing temperature of 71 °C to reduce non-specific products. After confirming the 332-bp libraries on a 2% agarose gel, each 100 μl library was cleaned with a 1× volume of SPRIselect beads to eliminate primers and eluted in Elution Buffer (Qiagen). Final PRISM sequencing libraries were quantified on the Agilent TapeStation 4200 with the High Sensitivity D1000 tape (Agilent) and diluted to 10 nM each. Samples with unique Illumina index combinations were pooled into 10 nM final libraries, with up to 96 samples potentially combined per pool due to 96 unique i5/i7 Illumina index combinations. Each pool also contained 10% molar ratio spike-in of PhiX to enhance sequence diversity. Data were analyzed by quantifying each unique 26 bp DNA barcode in the pool, which directly reflected the quantity of the corresponding cell line transduced with the barcode-containing construct in the PRISM mixture.

**Statistics and reproducibility**. Statistical analysis was performed using the GraphPad Prism 8 software. Two-group analysis was performed using unpaired two-tailed Student's $t$ test for normally distributed variables. One-way analysis of variance was applied where three groups were compared. Correlation analysis was performed using simple linear regression and then Pearson $r$ value was calculated to show the correlation coefficient. Experimental reproducibility was achieved by the following actions: (1) A total of 368 cell lines were used to demonstrate the good correlation between PRISM methodology and individual assays. (2) For each treatment condition (and time-course point), three biological replicates were included. All data are shown in dotplots to demonstrate data distribution and represent individual data points. Data are expressed as mean ± SEM.

**Reporting summary**. Further information on research design is available in the Nature Research Reporting Summary linked to this article.

## Data availability

This data set and all other source data generated or analyzed during this study are included in this published article (and its supplementary information file Supplementary Data 1). Full scan of western blot is shown in Supplementary Fig. 8. Additional details can be obtained from the corresponding author on reasonable request and proven by BMS legal department.

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

## Acknowledgements
We thank Louis Nguyenvu and Jin Jen for carrying out next-generation sequencing for the study.

## Author contributions
G.L. developed the original idea and designed barcode vectors. Y.X. established cell line library. C.S. and C.H optimized sequencing library preparation method. X.J. performed the screens. I.S.J. and K.W. performed bioinformatic analysis. G.L. and Y.X. interpreted data. M.R., N.B. and G.L. supervised the study.

## Competing interests
Y.X., X.J., I.S.J, C.S., K.W., M.R., N.B. and G.L. are or have been employees and equity holders at Bristol Myers Squibb. The other author claims no competing interests.
