## [Peer Review File · Communications Biology]

Reviewers' comments:

Reviewer #1 (Remarks to the Author):

This manuscript entitled "Genetic and Pharmacological Interrogation of Cancer Vulnerability Using a Multiplexed Cell Line Screening Platform" by Yifeng Xia with co-authors reports the development and optimization of a new high-throughput screening platform to systematically determine cancer cell sensitivity to pharmacological and genetic perturbations. The platform, named by authors next-generation PRISM or BMS-PRISM, leverages the power of two other platforms: PRISM and high-throughput CRISPR/Cas9 loss-of-function screen technology developed by Broad Institute. While PRISM allows pharmacological screens in mixtures of different cell lines, it cannot be applied to genetic perturbations. On the other hand, the established CRISPR/Cas9 methodology is limited by experiments in individual cell lines. To overcome these limitations, the authors applied the cell line barcoding approach implemented in the PRISM methodology to the CRISPR technology. Now this new approach can enable both pharmacological and genomic perturbations of individual genes in mixtures of up to 400 cell lines. The applicability of this method has been demonstrated using three different examples, including genetic and pharmacological perturbations of cell lines with and without alterations in EGFR, KRAS, and GSPT1 genes. A good agreement between genetic and pharmacological responses has been demonstrated. It was also shown that BMS-PRISM can recapitulate data previously obtained with PRISM and Crispr/Cas9 screens. Together, the reported data demonstrate the power and applicability of BMS-PRISM to rapidly discover new vulnerabilities in cancer. The manuscript is well organized, written clearly, and would be of interest to a broad readership of Communications Biology.

I would recommend this manuscript for publication in Communications Biology after addressing the minor comments provided below:

1. I assume "BMS" in BMS-PRISM states for Bristol Myers Squibb, though it should be clearly defined in the text.
2. The KRAS G12C experiments have been performed using the wild-type cells of all cancer types. Not all cancer types however have KRAS G12C and thus are not vulnerable to KRAS G12C knockout. It would be interesting to show the analysis performed for the subset of cancer types that do have KRAS G12C mutants.
3. Figure 2e and discussion of page 6 lines 101-103: It is unclear why the cell sensitivity to EGFR CRISPR knockout determined with the BMS-PRISM is compared with RNAi EGFR knockdown reported in the DepMap. It would be more appropriate to compare BMS-PRISM CRISPR data with the EGFR Crispr/Cas9 knockout data available from DepMap. The correlation plot between BMS-PRISM and DepMap CERES scores obtained for EGFR should be provided, and current Fig. 3f can be moved to SI.
4. The number of cell lines used in this study should be consistent throughout the text. For example, on page 4 it is indicated that "we have individually engineered more than 400 solid 61 tumor cell lines", on page 8 it is "CRISPR knockout screen with ~360 cell lines covering", and Figure 1 shows "~500 lines".
5. One of the major advantages of both the PRISM and DepMap project is their availability for the public. How the large scale BMS-PRISM data can be accessed? Should the readers expect that authors will apply the BMS-PRISM to profile cell response on the genome-wide scale and make that data available in the near future? Otherwise, the practical application of this new approach will remain very limited.

Reviewer #2 (Remarks to the Author):

In this interesting study, Yifeng Xia and collaborators describe the adaptation of the multiplexed cancer cell line screening platform PRISM to enable genetic perturbation screen of hundreds of cell lines by the combination of next-generation sequencing, CRISPR/Cas9, and barcode engineering. First of all, the authors have generated a collection of cancer cell lines by LV transduction each expressing spCas9, a blasticidin resistant gene, and a unique 26-bp DNA barcode. Secondly, they have developed an NGS protocol for quantification of the percentage of each barcode in a pooled cell mixture to have a representation of the relative cell number of the corresponding cell line in a mixed pool. To optimize the screening protocol, the authors have used 20 NSCLC cell lines wt or EGFR-mutant to establish their EGFR dependence. To that aim, the authors include 293T control cells into the mix as internal references. The authors have used the relative barcode abundance of 293T in the cell line mix to calculate the barcode representation of each cell line in the same PRISM mix. As a second validation of their technology, the authors compared G12C KRAS mutant vs non-G12C cell lines

Major points:

- 1.- Which MOI has been used to generate the collection of cancer cell lines? This is a key step the number of constructs integrated into each could be different and therefore modify the relative cell number count in the pooled cells culture. The authors should give information about the analysis of the number of constructs integrated into each cell line included in the study.
- 2.- Proliferation rate of 293T and NSCLC cancer cell lines should be compared. The authors use the relative 293T barcode abundance to calculate the barcode representation of each cell line. To perform this calculation, it is important to take into account the proliferation rate of each cell line compared to 293T cells and the number of constructs integrated into 293T and the NSCLC cell lines.
- 3.- The authors report differences in gene essentiality in the pooled PRISM screen and indicate that this could be attributed to autocrine and paracrine effects in the culture. The authors state that further investigations are needed. These investigations must be included in the present study.
- 4.- The authors use a "non-targeting" sgRNA as a negative control throughout their study, which is not sufficient to control for unspecific effects of Cas9 DNA cleavage and repair. It is well known that Cas9-induced double-strand breaks trigger DNA damage response programs that can have unspecific anti-tumour effects, and to rule out such effects the appropriate control would be to use a sgRNAs that effectively cleave DNA at another locus (ideally in a safe harbour place).
- 5.- If it is needed to include the 293T cells into the pooled mix of cell lines it is important to determine beforehand the effect of KO of a specific gene in 293T cells. This should be discussed in the manuscript.

Minor points:

- Fig 2a label "PRSIM" should be "PRISM"
- Fig2b, it is difficult to distinguish between the blue and green dots/squares.
- Fig 3 labels "Kras" should be "KRAS" Human gene symbols are italicized, all letters are in upper case

Reviewer #3 (Remarks to the Author):

The manuscript by Xia et al. describes the utility of a multiplexed cell line screening platform "PRISM" with CRISPR screening. This methodology is very interesting and carefully designed. The manuscript is well written. However, is it for validation experiments? "PRISM" and CRISPR screening are both well-established. New findings or applications of this method are expected.

Minor point:

In Figure 2d, the x-axis labeling should be capitalized.

We want to thank all three Reviewers for their very constructive suggestions which have greatly improved the quality of our manuscript. We have carefully revised our manuscript accordingly. In this rebuttal letter, we believe we have addressed most of the Reviewers' concerns. Please see our point-to-point response below.

Referee expertise:

Referee #1: high-throughput screening technologies, cancer therapeutic discovery

Referee #2: CRISPR/Cas for cancer therapy

Referee #3: CRISPR/Cas for drug screening

Reviewers' comments:

Reviewer #1 (Remarks to the Author):

This manuscript entitled "Genetic and Pharmacological Interrogation of Cancer Vulnerability Using a Multiplexed Cell Line Screening Platform" by Yifeng Xia with co-authors reports the development and optimization of a new high-throughput screening platform to systematically determine cancer cell sensitivity to pharmacological and genetic perturbations. The platform, named by authors next-generation PRISM or BMS-PRISM, leverages the power of two other platforms: PRISM and high-throughput CRISPR/Cas9 loss-of-function screen technology developed by Broad Institute. While PRISM allows pharmacological screens in mixtures of different cell lines, it cannot be applied to genetic permutations. On the other hand, the established CRISPR/Cas9 methodology is limited by experiments in individual cell lines. To overcome these limitations, the authors applied the cell line barcoding approach implemented in the PRISM methodology to the CRISPR technology. Now this new approach can enable both pharmacological and genomic perturbations of individual genes in mixtures of up to 400 cell lines. The applicability of this method has been demonstrated using three different examples, including genetic and pharmacological perturbations of cell lines with and without alterations in EGFR, KRAS, and GSPT1 genes. A good agreement between genetic and pharmacological responses has been demonstrated. It was also shown that BMS-PRISM can recapitulate data previously obtained with PRISM and Crispr/Cas9 screens. Together, the reported data demonstrate the power and applicability of BMS-PRISM to rapidly discover new vulnerabilities in cancer. The manuscript is well organized, written clearly, and would be of interest to a broad readership of Communications Biology.

I would recommend this manuscript for publication in Communications Biology after addressing the minor comments provided below:

1. I assume "BMS" in BMS-PRISM states for Bristol Myers Squibb, though it should be clearly defined in the text.

Thanks to the Reviewer for pointing this out, we have updated accordingly in the manuscript where "BMS-PRISM" first appears.

2. *The KRAS G12C experiments have been performed using the wild-type cells of all cancer types. Not all cancer types however have KRAS G12C and thus are not vulnerable to KRAS G12C knockout. It would be interesting to show the analysis performed for the subset of cancer types that do have KRAS G12C mutants.*

KRAS dependency has been carefully evaluated in many studies and consistent data showed that some KRAS mutant cell lines are not KRAS-dependent in 2D culture system (both G12C inhibitor and RNAi, also discussed in the manuscript). This conclusion holds true in KRAS G12C mutant cell lines in our PRISM collection, as we showed in both PRISM and individual assays (Fig. 3d).

Per Reviewer's suggestion, we added a plot of all 17 KRAS G12C cell lines' response to KRAS CRISPR KO or G12C inhibitor AMG-510 in Supplementary Fig.S5a. As expected, not all KRAS G12C lines responded well to KRAS inhibition, however, their sensitivity to KRAS KO and AMG-510 is quite consistent ($r=0.6951$).

3. *Figure 2e and discussion of page 6 lines 101-103: It is unclear why the cell sensitivity to EGFR CRISPR knockout determined with the BMS-PRISM is compared with RNAi EGFR knockdown reported in the DepMap. It would be more appropriate to compare BMS-PRISM CRISPR data with the EGFR Crispr/Cas9 knockout data available from DepMap. The correlation plot between BMS-PRISM and DepMap CERES scores obtained for EGFR should be provided, and current Fig. 3f can be moved to SI.*

We totally agree with the Reviewer that a correlation plot between BMS-PRISM and DepMap CRISPR (CERES) scores should be a more relevant comparison. We thus updated Fig. 2f accordingly with DepMap CRISPR data and moved RNAi data to Supplementary Fig. S3c, while the overall conclusion remained unchanged. The reason we presented correlation plot between BMS-PRISM and DepMap RNAi (DEMETER) in the original manuscript was that the correlation score of BMS-PRISM vs. DepMap RNAi ($r=0.6813$) was slightly higher than that of BMS-PRISM vs. DepMap CRISPR ($r=0.5098$). We believe this was due to the special normalization algorithm DepMap CERES scores applied. More specifically, CERES method regressed out copy number difference, while the EGFR copy numbers of those 20 NSCLC lines do vary largely, as shown in Supplementary Table S1.

For consistency, we also updated KRAS correlation plot in Fig. 3e with BMS-PRISM vs. DepMap CRISPR.

4. *The number of cell lines used in this study should be consistent throughout the text. For example, on page 4 it is indicated that "we have individually engineered more than 400 solid 61 tumor cell lines", on page 8 it is "CRISPR knockout screen with ~360 cell lines covering", and Figure 1 shows "~500 lines".*

We thank the Reviewer for pointing this inconsistency out. We have updated the numbers accordingly in the manuscript as well as in the figures. To clarify, we have engineered more than 400 solid tumors lines and only those having Cas9 editing efficiency higher than 85% were included in the PRISM mix (current version = 368 lines).

5. *One of the major advantages of both the PRISM and DepMap project is their availability for the public. How the large scale BMS-PRISM data can be accessed? Should the readers expect that authors will apply the BMS-PRISM to profile cell response on the genome-wide scale and make that data available in the near future? Otherwise, the practical application of this new approach will remain very limited.*

We fully understand the Reviewer's concerns regarding to the public availability of BMS-PRISM platform. We are aligned on the importance of sharing our research with the wider community and will make all plasmid sequences and detailed experimental methods readily available. However, we are unable to share the entire engineered cell line collection owing to our company's internal policies and the terms of various Material Transfer Agreements with the providers of the cell lines we used (ATCC etc). Besides, the original purpose of developing BMS-PRISM was to facilitate our internal drug development activities, therefore we will only apply the platform to screen genes and compounds of our own interests instead of the whole genome. We apologize that we will not be able to share our screen data in the near future based on these considerations.

Reviewer #2 (Remarks to the Author):

In this interesting study, Yifeng Xia and collaborators describe the adaptation of the multiplexed cancer cell line screening platform PRISM to enable genetic perturbation screen of hundreds of cell lines by the combination of next-generation sequencing, CRISPR/Cas9, and barcode engineering. First of all, the authors have generated a collection of cancer cell lines by LV transduction each expressing spCas9, a blasticidin resistant gene, and a unique 26-bp DNA barcode. Secondly, they have developed an NGS protocol for quantification of the percentage of each barcode in a pooled cell mixture to have a representation of the relative cell number of the corresponding cell line in a mixed pool. To optimize the screening protocol, the authors have used 20 NSCLC cell lines wt or EGFR-mutant to establish their EGFR dependence. To that aim, the authors include 293T control cells into the mix as internal references. The authors have used the relative barcode abundance of 293T in the cell line mix to calculate the barcode representation of each cell line in the same PRISM mix. As a second validation of their technology, the authors compared G12C KRAS mutant vs non-G12C cell lines

Major points:

1.- Which MOI has been used to generate the collection of cancer cell lines? This is a key step the number of constructs integrated into each could be different and therefore modify the relative cell number count in the pooled cells culture. The authors should give information about the analysis of the number of constructs integrated into each cell line included in the study.

We totally agree with the Reviewer that MOI is a critical factor when generating these barcoded cell lines. Not like typical CRISPR screen where MOI needs to be lower than 1 to ensure that each cell gets no more than one viral particle. In the case of our barcode-labeling, we used approximately MOI=8 to infect cells so that the cells have enough Cas9 protein expression (described in Supplementary Methods). Besides, we only enriched those infected cells with blasticidin selection to avoid potential clonal bias. Therefore, the barcode copy number of each cell line in the PRISM collection varies and we do not adjust the input cell number to ensure exact same abundance of each barcode when initiating the screen. Instead, we have a DMSO (or sgNT) control for each time point as a reference. All the treatment group results are finally compared to that of the control sample to calculate the compound (or gene) effect.

This information has been added to the Discussion part of the revised manuscript.

2.- Proliferation rate of 293T and NSCLC cancer cell lines should be compared. The authors use the relative 293T barcode abundance to calculate the barcode representation of each cell line. To perform this calculation, it is important to take into account the proliferation rate of each cell line compared to 293T

cells and the number of constructs integrated into 293T and the NSCLC cell lines.

We agree with the Reviewer that the proliferation rate of different cell lines needs to be considered in the PRISM screen. Similar to the point #1 (MOI and barcode copy number) raised by the Reviewer, the difference of cell line proliferation rate will also affect the relative abundance of each barcode during the screen. For example in the NSCLC pilot screen, 293T and NCI-H460 proliferated much faster than other lines. Altogether, several factors may contribute to the relative abundance besides to sensitivity to the testing compound: copy number of barcode integrated into the cell, initial input of cell number, ability to attach to the culture plate, proliferation rate, etc. To solve this problem, we have a DMSO (or sgNT) control for each time point as a reference. All the treatment group results are finally compared to that of the control sample to calculate the compound (or gene) effect.

This information has been added to the Discussion part of the revised manuscript.

3.- The authors report differences in gene essentiality in the pooled PRISM screen and indicate that this could be attributed to autocrine and paracrine effects in the culture. The authors state that further investigations are needed. These investigations must be included in the present study.

It is the nature of the PRISM system that paracrine effects cannot be correctly compensated and may interfere the interpretation of data. However, in our limited cell lines tested, the PRISM data correlated well with the individual assay results (Fig. 2c, 2e, 3d).

Broad PRISM reported on their website that “When testing ~100 cell lines to compare pooled/unpooled growth rates, we found one cell line causing a paracrine effect...”. So we believe total number of cell lines that may cause paracrine effect is very small.

4.- The authors use a “non-targeting” sgRNA as a negative control throughout their study, which is not sufficient to control for unspecific effects of Cas9 DNA cleavage and repair. It is well known that Cas9-induced double-strand breaks trigger DNA damage response programs that can have unspecific anti-tumour effects, and to rule out such effects the appropriate control would be to use a sgRNAs that effectively cleave DNA at another locus (ideally in a safe harbour place).

We fully agree with the Reviewer that CRISPR editing perturbation may induce unspecific DNA damage response and impair cell fitness. Therefore, we always have both non-targeting sgRNA (eg. NT; does not match any genomic sequence) and non-coding sgRNA (eg. NC, matches HBG1 gene intron sequence; HBG1 is normally expressed in the fetal liver, spleen and bone marrow, therefore targeting its intron sequence in cancer cell lines is expected to have minimal on-target effect) controls in our CRISPR screen. Our early individual test results indicated that either sgNC or sgNT had little impact on cell fitness. Here we also provided a plot of relative cell percentage of each cell line in the BMS-PRISM upon the infection of sgNT and sgNC. Again, there was little difference observed ($r=0.9629$), therefore we are confident that either sgNT or sgNC can be used for CRISPR PRISM screen normalization in most scenarios. This result has also been added to Discussion and Supplementary Fig. S5b.

5.- If it is needed to include the 293T cells into the pooled mix of cell lines it is important to determine beforehand the effect of KO of a specific gene in 293T cells. This should be discussed in the manuscript. We understand that 293T may also respond to CRISPR perturbation therefore it could not be a true negative control for reads normalization. To address this problem, we actually have four engineered 293T lines with unique barcodes in the screen pool (two 293T lines with active Cas9 gene and two 293T lines with truncated/dead Cas9 gene), and we use the latter for normalization in the CRISPR screen. This information was in the original manuscript and Supplementary Methods.

Minor points:

- Fig 2a label “PRSIM” should be “PRISM”
 - Fig2b, it is difficult to distinguish between the blue and green dots/squares.
 - Fig 3 labels “Kras” should be “KRAS” Human gene symbols are italicized, all letters are in upper case
- We thank the Reviewer for these suggestions. We have updated accordingly.

Reviewer #3 (Remarks to the Author):

The manuscript by Xia et al. describes the utility of a multiplexed cell line screening platform “PRISM” with CRISPR screening. This methodology is very interesting and carefully designed. The manuscript is well written. However, is it for validation experiments? “PRISM” and CRISPR screening are both well-established. New findings or applications of this method are expected.

We thank the Reviewer for very positive feedback. The major application for BMS-PRISM platform is to support our internal early drug development. We apologize that we could not release additional screen result of proprietary compounds or potential target genes due to company’s intellectual policy.

Minor point:

In Figure 2d, the x-axis labeling should be capitalized.

We thank the Reviewer for the suggestion. We believe what the Reviewer meant was the “y-axis labeling”, so we have updated to “PRISM AUC”.

REVIEWERS' COMMENTS:

Reviewer #1 (Remarks to the Author):

The authors have addressed all major issues and have revised the manuscript accordingly. I do not have further comments and I recommend the manuscript be accepted for publication.

Reviewer #2 (Remarks to the Author):

In their revised manuscript, the authors address a number of critical points raised by the reviewers. However, one of my major concerns (very fundamental) was not addressed. I agree with the authors that it is the nature of the PRISM system that paracrine effects cannot be correctly compensated and may interfere with the interpretation of data. However, as the authors themselves state in the manuscript "further investigations are needed" to address this issue. In my first revision, I asked to go deeper and develop some investigations. The authors have not performed any experiment to address the comments. And only indicates in their response that they "believe" that the total number of cell lines that may cause the paracrine effect is very small. This should be demonstrated in the present study.

EDITOR'S COMMENTS:

We ask that you kindly address the final point from Reviewer 2 regarding paracrine effects. If further data cannot be provided at this stage, we ask that you include a statement on this limitation in the discussion.

We thank the Editor for understanding. The original purpose of developing BMS-PRISM platform was to examine gene dependency and drug response in a high-throughput manner. After the quick screen, we still need to further evaluate the cell line-of-interest individually, both *in vitro* and *in vivo*. It is the nature of PRISM platform (both ours and Broad Institute's) that we couldn't avoid the potential paracrine effect, although the percentage of cell lines having this concern is low according to our current results.

As the Editor suggested, we have added extensive discussion in the revised manuscript. It reads:

"One of the major differences between the PRISM screen and individual assays that may introduce inconsistency is the cell culture condition. First, nearly 400 cell lines were mixed together and cultured in a single pool for 5-10 days in the PRISM screen settings, and therefore paracrine signal(s) may affect the dependence on a gene of interest in certain cell populations. Although it is conceivable that the paracrine effects of genes involved in regulation of cell proliferation signals, such as EGFR and KRAS, should become more dominant in the PRISM screen, we observed minimal discrepancy between the PRISM screens and their individual assays when EGFR and KRAS were tested in a focused screen format and full-library screen format, respectively. We reasoned that the percentage of cell lines that could potentially affect the cell fitness of others via a paracrine effect controlled by any gene of interest in the PRISM screen is probably below 5%, thus this paracrine effect, if existed, should not affect the overall scoring of gene dependency in the PRISM screen. This hypothesis is strongly supported by our data shown in this study as well as in the PRISM screen result published by the Broad Institute^{8,9}. On the other hand, given the limited cell number of each cell line (roughly 1000 cells/per line) in the PRISM screen pool, the autocrine effects observed in cell lines when cultured as a homogenous population may not be detected in the PRISM screen, which could lead to differential addiction to genes affecting cell growth via an autocrine signal. Lastly, cell seeding density is another plausible factor affecting the readout of gene dependency in the PRISM screen. Notably, cell density is relatively higher in the PRISM screen than in standard CTG or CRISPR competition assays, so cell-cell and cell-matrix interactions may vary slightly from traditional cell culture conditions. Maintaining the cells in a monolayer in the PRISM screen helps to overcome this limitation, since cells may acquire some features of 3D culture conditions once they outgrow the monolayer. For example, since we were able to maintain cells in a monolayer in the PRISM setting, we did not observe increased dependency for KRAS of any cell lines, which has previously been shown under 3D culture conditions. Nevertheless, further investigation is needed to assess the difference in cell culture condition on gene dependency especially in cell lines showing differential addiction to a given gene of interest. Moreover, since tumor cells in a human body typically grow in a nutrient-limited microenvironment where complex autocrine and paracrine cell-cell and/or cell-matrix interactions control the tumor growth and survival, the true dependence on any candidate gene needs to be validated in a panel of cell line or patient-derived xenograft models *in vivo*."

We therefore invite you to revise your paper one last time to address the remaining concerns of our reviewers. At the same time we ask that you edit your manuscript to comply with our format requirements and to maximise the accessibility and therefore the impact of your work, including providing source data and other formatting requests.

Manuscript has been updated as requested.

REVIEWERS' COMMENTS:

Reviewer #1 (Remarks to the Author):

The authors have addressed all major issues and have revised the manuscript accordingly. I do not have further comments and I recommend the manuscript be accepted for publication.

We thank the Reviewer for the recommendation.

Reviewer #2 (Remarks to the Author):

In their revised manuscript, the authors address a number of critical points raised by the reviewers. However, one of my major concerns (very fundamental) was not addressed. I agree with the authors that it is the nature of the PRISM system that paracrine effects cannot be correctly compensated and may interfere with the interpretation of data. However, as the authors themselves state in the manuscript "further investigations are needed" to address this issue. In my first revision, I asked to go deeper and develop some investigations. The authors have not performed any experiment to address the comments. And only indicates in their response that they "believe" that the total number of cell lines that may cause the paracrine effect is very small. This should be demonstrated in the present study.

We apologize that we couldn't provide further investigation data on "paracrine effect" at this time .Please see our response to the Editor above.